# Detection of Microorganisms Causing Human Respiratory Infection Using One-Tube Multiplex PCR

**DOI:** 10.3390/idr17040093

**Published:** 2025-08-04

**Authors:** Isabela L. Lima, Adriana F. Neves, Robson J. Oliveira-Júnior, Lorrayne C. M. G. Honório, Vitória O. Arruda, Juliana A. São Julião, Luiz Ricardo Goulart Filho, Vivian Alonso-Goulart

**Affiliations:** 1Instituto de Biotecnologia, Universidade Federal de Uberlândia, Uberlândia 38405-302, Minas Gerais, Brazil; isabela.lemos@ufu.br (I.L.L.); oliveirajunior@ufu.br (R.J.O.-J.); lorraynegarcia@ufu.br (L.C.M.G.H.); 2Instituto de Biotecnologia, Universidade Federal de Catalão, Catalão 75704-020, Goiás, Brazil; adriana_freitas_neves@ufcat.edu.br (A.F.N.); vitoriaoliveira@discente.ufg.br (V.O.A.); 3Technical Department, BioGenetics Tecnologia Molecular Ltda, Uberlândia 38400-446, Minas Gerais, Brazil; juliana@biogenetics.com.br

**Keywords:** diagnosis, respiratory syndromes, RT-PCR, SARS-CoV-2

## Abstract

**Background/Objectives:** Due to the significant overlap in symptoms between COVID-19 and other respiratory infections, a multiplex PCR-based platform was developed to simultaneously detect 22 respiratory pathogens. Target sequences were retrieved from the GenBank database and aligned using Clustal Omega 2.1 to identify conserved regions prioritized for primer design. Primers were designed using Primer Express^®^ 3.0.1 and evaluated in Primer Explorer to ensure specificity and minimize secondary structures. A multiplex strategy organized primers into three groups, each labeled with distinct fluorophores (FAM, VIC, or NED), allowing for detection by conventional PCR or capillary electrophoresis (CE). **Methods:** After reverse transcription for RNA targets, amplification was performed in a single-tube reaction. A total of 340 clinical samples—nasopharyngeal and saliva swabs—were collected from patients, during the COVID-19 pandemic period. The automated analysis of electropherograms enabled precise pathogen identification. **Results:** Of the samples analyzed, 57.1% tested negative for all pathogens. SARS-CoV-2 was the most frequently detected pathogen (29%), followed by enterovirus (6.5%). Positive results were detected in both nasopharyngeal and saliva swabs, with SARS-CoV-2 predominating in saliva samples. **Conclusions:** This single-tube multiplex PCR-CE assay represents a cost-effective and robust approach for comprehensive respiratory pathogen detection. It enables rapid and simultaneous diagnosis, facilitating targeted treatment strategies and improved patient outcomes.

## 1. Introduction

Human respiratory infections present with diverse respiratory symptoms and varying levels of severity, especially in young, elderly, immunocompromised, and other high-risk groups [1]. These infections represent a significant global health challenge, contributing to morbidity, mortality, and economic burden worldwide [2,3]. The most common clinical syndrome is the common cold; however, in individuals with chronic respiratory conditions or compromised immune systems, these infections can result in severe clinical complications. While viruses are the primary causative agents, bacterial pathogens also contribute to respiratory infections [4,5,6,7]. Diseases caused by respiratory pathogens often present with a range of common and nonspecific symptoms, highlighting the importance of diagnostic tests capable of identifying specific etiological agents [8,9].

Since the emergence of COVID-19, reports of other types of acute respiratory infections have increased, creating challenges to healthcare systems. This rise is due to similar symptoms shared by SARS-CoV-2 and other respiratory viruses, emphasizing the need for accurate identification [10,11]. Polymerase chain reaction (PCR), including its variations like real-time PCR (qPCR), remains the gold standard for diagnosing respiratory illnesses caused by SARS-CoV-2 and other pathogens [12,13,14]. When combined with automated fragments analysis, PCR can detect multiple genetic materials from different pathogens at once. This improvement enhances diagnostic accuracy and helps reduce unnecessary antibiotic use for non-bacterial infections or cases that present severe respiratory symptoms [8].

Traditional diagnostic methods face challenges like difficult cultures, long turnaround times, and the need to assume which pathogens are involved [15]. Given the limitations of current technologies, this study aimed to develop a PCR-multiplex (MPCR) platform combined with capillary electrophoresis (CE) for the simultaneous detection of 22 pathogens associated with respiratory infection. This MPCR-CE platform utilizes pathogen-specific primers to enable the differential diagnoses of respiratory infections, including SARS-CoV-2, within a single-tube assay.

## 2. Materials and Methods

### 2.1. Saliva and Nasopharyngeal Swab Human Samples Collection

This study received ethical approval from the Human Research Ethics Committee of the Universidade Federal de Uberlândia (approval number CAAE: 30848620.1.0000.5704 and 69764423.7.3002.0164). All participants provided written informed consent before participation. All experiments were performed according to national regulations. Samples were obtained through a partnership with a private clinical laboratory from September 2020 to September 2021. All patients who requested molecular testing for COVID-19 were invited to participate in the study. Those who agreed followed the laboratory’s standard collection procedures for the molecular detection of SARS-CoV-2, which included nasopharyngeal swabs or the collection of 3 to 5 mL of saliva (289 nasopharyngeal samples and 51 saliva samples). The sample type and collection methods were already standardized for SARS-CoV-2 molecular testing. Samples were excluded if they were received with a dry swab or with less than 3 mL of saliva. Due to the context of the pandemic, the study population was characterized by the presence of various respiratory symptoms and being over 18 years of age. Any patient presenting one or more respiratory symptoms resembling flu-like or cold-like conditions, or who had contact with a confirmed COVID-19 case, was included in the study group. The nasopharyngeal swab or saliva samples from 340 patients were collected for routine clinical testing. Aliquots of samples were stored at −80 °C until analysis.

### 2.2. Targets and Primer Design

The diagnostic platform was developed to detect a broad panel of respiratory pathogens, including adenovirus, enterovirus, influenza A virus (with H1N1 subtype), influenza B virus, human parainfluenza viruses 1–4, four human coronaviruses (NL63, 229E, HKU1, and OC43), human parechovirus, respiratory syncytial viruses A and B, human metapneumoviruses A and B, *Mycoplasma pneumoniae*, human rhinovirus, human bocavirus, and SARS-CoV-2. Target sequences were retrieved from the NCBI GenBank database and aligned using Clustal Omega 2.1 (European Molecular Biology Laboratory, Cambridge, UK) [4] to identify regions of high sequence similarity across strains, which were then prioritized for primer design to ensure broad detection capability. These conserved regions were selected to maximize the primers’ ability to recognize diverse genetic variants of each pathogen. Oligonucleotides were designed using Primer Express^®^ 3.0.1 (ThermoFisher Scientific, Waltham, MA, USA), with each primer pair tailored to amplify sequences of approximately 20 base pairs, featuring an annealing temperature near 60 °C and a GC content between 40 and 60% to balance stability and specificity. Following initial design, the primers were analyzed in Primer Explorer (Version 5, Eiken Chemical Co., Tokyo, Japan) to evaluate potential secondary structures, including hairpins and primer-dimers, and to confirm optimal annealing temperatures. This step was critical to minimize off-target interactions and ensure robust amplification efficiency. To address overlapping amplicon sizes, a multiplexing strategy was implemented, organizing the primers into three groups, each labeled with a distinct fluorophore (VIC, FAM, or NED). Within each group, amplicon sizes were designed to differ by at least 20 base pairs, enabling clear discrimination during capillary electrophoresis. By leveraging conserved genomic regions for primer design and combining fluorophore labeling with precise amplicon sizing, the platform achieved the simultaneous detection and differentiation of all 22 targets in a single, high-throughput assay.

### 2.3. Plasmid Engineering to Validate Oligonucleotides

Plasmids designed for use as positive controls feature distinct sequences from the genome of each species, serving as target products for amplification. They are employed as positive controls for comparison with patient samples. In the same way that the oligonucleotides were categorized into three groups based on the fluorophore utilized, the plasmids were also grouped into three, with each containing the inserts of interest within its respective category primers labeled with fluorophore. Then, the plasmid DNA was engineered. For each pair of oligonucleotides, an amplification reaction was performed using the respective plasmid containing the insert of interest to validate the specificity of the primers. This first amplification reaction was subjected to electrophoresis in a 3% agarose gel. In the second step, amplification reactions were performed for each group. In other words, each reaction contained all primer pairs from its group. This second reaction was validated using CE on the Applied Biosciences 3500xL^®^ Genetic Analyser (Manufactured by Thermo Fisher Scientific, Paisley, UK).

### 2.4. One-Tube PCR Multiplex

Nucleic acid (dependent on the microorganism) was extracted from samples from human nasopharyngeal swabs or saliva. The samples were homogenized and subjected to genetic material extraction with the MagMax™ Viral/Pathogen Nucleic Acid Isolation kit (Version I, Applied Biosystem, Waltham, MA, USA), containing magnetic beads. As most of the pathogens in the assay have RNA as genetic material, a reverse transcriptase reaction was performed using random primers (RT-PCR). After converting RNA into cDNA, the polymerase chain reaction (PCR) was carried out, which contains 22 pairs of primers divided into three groups, where each group was marked with a fluorophore (VIC, FAM, or NED), which amplifies specific regions of the 22 microorganisms (21 viruses and 1 bacteria) target. The first reaction was the conversion of RNA into cDNA, which consists of three temperature cycles ranging from 37 °C to 70 °C. The reaction components were as follows: 1 × Enzyme Buffer, 40 u of M-MLV Reverse Transcriptase enzyme, 106 pmol of random primers, and 100 mM of each dNTP, completing the volume to a final 25 μL of reaction with water treated with diethyl pyrocarbonate (DEPC). The second reaction, which consisted of PCR (multiplex), used 5 µL of cDNA from the first reaction, which was amplified simultaneously by 22 pairs of primers in a single tube, with several temperature cycles ranging from 59 °C to 94 °C. The reaction components were as follows: 10 × Taq buffer, 1.5 U of Taq DNA polymerase, 100 μM of each dNTP, 6 pmol of each specific primer, and 0.05 mM of MgCl2, completing the reaction volume to 25 μL finals with ultrapure water.

Then, denaturing CE was performed on the Applied Biosciences 3500^®^ sequencer (Manufactured by Thermo Fisher Scientific, Paisley, UK). The amplified products were diluted in water 1:20. Then, 1 µL of the diluted product from each sample to be tested went through a denaturation process with formamide at a temperature of 94 °C for 3 min and was added to the high-density, dye-labeled size standard for reproducible sizing for fragment analysis data (GeneScan™ 600 LIZ^®^). After this process, the sample underwent a thermal shock, was exposed to a temperature of 4 °C, and was injected into the Applied Bioscience 3500^®^ sequencer to perform capillary electrophoresis. Fluorescence and molecular weight readings were performed to identify target peaks present in the samples, with the results showing positive peaks in the expected region(s) of the electropherogram. Analyses were performed using GeneMaper^®^ software (Version: 1.6, Applied Biosystems, Foster City, CA, USA) after capillary electrophoresis. The fluorophores were read at their respective wavelengths—VIC (538 nm), NED (546 nm), or FAM (494 nm)—and the targets were detected by fluorescence and molecular weight. The analysis provided qualitative results, that is, positive or negative. Positive results are those that detect one or more targets, and negative results are those that do not detect any of the targets.

## 3. Results

### 3.1. Design of Fluorescently Labeled Oligonucleotides

The targets chosen for this panel were grouped based different fluorophores:Fluorophore FAM: enterovirus (EV), influenza A virus (IAV), human parainfluenza virus 4 (HPIV-4), HCoV-Nl63 (Cor-63), parainfluenza virus 2 (HPIV-2), influenza virus B (IVB), HCoV-229e (Cor-229), HCoV Hku1 (HKU).Fluorophore VIC: human parechovirus (HPeV), respiratory syncytial virus A (RSV-A), human parainfluenza virus 3 (HPIV-3), human metapneumovirus B (HMPV-B), *Mycoplasma pneumoniae* (Mpneu), human parainfluenza virus 1 (HPIV-1).Fluorophore NED: human metapneumovirus A (HMPVA), HCoV-OC43 (Cor-43), adenovirus (AdV), influenza A virus (H1N1), human rhinovirus (RV), human bocavirus (HBoV), SARS-CoV-2 (SARS-CoV-2N2), respiratory syncytial virus B (RSV-B).

All target sequences were retrieved from the NCBI GenBank database. The oligonucleotides were designed, and when necessary, the sequences were degenerated. The annealing temperature was approximately 60 °C to facilitate the optimization process. The products with the purpose of multiplex PCR (MPCR) exhibit distinct sizes, enabling precise identification based on both fragment size and fluorescence intensity during capillary electrophoresis (CE) analysis. Table 1 lists the target pathogens along with their respective specific oligonucleotides (forward and reverse).

Because the expected amplicons for the different targets were similar in size, they were divided into three groups, each labeled with a different fluorophore to enable simultaneous detection. This approach allowed for distinguishing amplicons with small differences in base pairs, especially in cases of co-infections. Table 2 summarizes how the targets were grouped by fluorophore (FAM, VIC, NED), according to the filters available on the CE equipment.

PUC18 plasmids were designed with distinct sequences from each species’ genome to serve as positive controls and target products for amplification when compared with patient samples.

### 3.2. Validation of Oligonucleotides with Engineered Plasmid

The panel grouping for each pathogen causing respiratory disease is shown in Figure 1. Each target is represented by its amplicon corresponding to the expected molecular weight, after singleplex PCR using plasmid DNA as control of the reaction and validating the set of primers. The results confirm that the oligonucleotides were specific for each target detection.

We also observed the size scale between the targets in each group; the primers were able to specifically amplify targets with a minimum difference of 20 bp. The PUC18PDR-Vic plasmid shows HPeV, RSV-A, HPIV-1, HPIV-3, HMPV-B, and Mpneu (Figure 1A); the PUC18 PDR-Fam plasmid shows EV, IAV, HPIV-4, Cor-63, HPIV-2, IBV, Cor-229, and HKU (Figure 1B); and the PUCPDR-Ned plasmid shows the HMPVA, Cor-43, AdV, H1N1, RV, HBoV, SARS-CoV-2N2 and RSV-B sequence amplification (Figure 1C).

Capillary electrophoresis allowed for the detection of all targets within their expected regions (BINS), which corresponds to the number of base pairs in the amplicons as identified by fluorescence. Each PCR reaction contained 10 copies of each plasmid, and the amplified products were diluted 20 to 30 times before being injected into the capillary system. Clear peaks were observed for each target within their respective BINS, with sufficient fluorescence intensity for reliable detection, while also minimizing cross-channel contamination (Figure 2).

### 3.3. Multiplex PCR One Tube with 44 Oligos

After validating the oligonucleotides using plasmids as positive controls, nasopharyngeal swab and saliva samples were collected from patients exhibiting respiratory symptoms. To begin validation with real patient samples—especially given the SARS-CoV-2 pandemic—SARS-CoV-2 was chosen as the first pathogen to test in the multiplex panel. All primers for this pathogen were combined in a single tube, using samples confirmed as positive by multiplex PCR (MPCR). In addition to the primers designed for the panel, a secondary target for SARS-CoV-2 was included as a positive control (PC), which had been tested previously. As illustrated in Figure 3, amplification of SARS-CoV-2 (PC) (228-bp) and SARS-CoV-2 N2 (310 bp) showed the expected peaks in CE Figure 3A, from a patient with a positive SARS-CoV-2 diagnosis. Figure 3B shows results for 11 patient samples analyzed on a 3% agarose gel, and three samples showed the two expected bands for SARS-CoV-2.

The 340 samples analyzed using MPCR-CE for the differential diagnosis of 22 respiratory infections presented the following results: 57.1% negative, 29% positive for SARS-CoV-2, 6.5% positive for enterovirus, 3.5% positive for HCoV-229E, 0.6% positive for respiratory syncytial virus A, 1.5% positive for HCoV-OC43, and 1.8% inconclusive. Positive results were detected in both nasopharyngeal and saliva swabs, with SARS-CoV-2 predominating in saliva samples.

These results demonstrate that the multiplex PCR-based method, MPCR-CE, can reliably detect and distinguish respiratory infections with high throughput—up to 96 samples per batch—and a quick turnaround time of approximately 210 min. It is also capable of differentiating targets of similar sizes. Figure 4 provides examples of other viruses identified by the panel in samples from patients exhibiting respiratory symptoms.

## 4. Discussion

Since the start of the COVID-19 pandemic, there has been a significant increase in reports of other acute respiratory infections, highlighting the urgent need for the comprehensive detection of various respiratory viruses [8]. The pandemic has profoundly altered the epidemiology of respiratory viruses. Given these shifts, there is an urgent demand for molecular assays that can simultaneously detect multiple respiratory viruses and SARS-CoV-2. In this study, we developed and validated a multiplex PCR assay combined with capillary electrophoresis to identify 22 respiratory pathogens, including SARS-CoV-2, all in a single reaction tube. We designed the primers using Primer Express^®^ 3.0.1, but in the future, we can look forward to AI integrationwhich utilized the application of neural networks for primer design and the prediction of PCR results [16].

The multiplex PCR-based diagnostic test proposed here effectively identified various respiratory infections. Following COVID-19, which was detected in 29% of samples, the most prevalent pathogen was enterovirus. Enterovirus is commonly associated with the common cold [7]. Influenza, a respiratory illness caused by influenza viruses (A and B), is known for its seasonal and annual epidemics, with pandemic outbreaks typically linked to specific influenza A strains [7]. These factors might explain why we did not detect the influenza virus in our samples. Additionally, the samples were collected during the COVID-19 pandemic, which could have influenced the results. Additionally, a small percentage of samples tested positive for other coronaviruses (HCoV-229E and HCoV-OC43), which generally cause mild upper respiratory symptoms and account for up to 30% of common cold cases in adults [17]. The samples were collected and tested after the highest peaks in COVID-19 cases, which explains the number of negative tests.

Assays for the simultaneous diagnosis of some respiratory pathogens have been described previously; however, these reactions are typically conducted in multiple tubes. Studies in literature developed primers capable of detecting various types of coronaviruses, which were subsequently divided into three tubes based on their fluorophores [18]. Similarly, other study developed a diagnostic test for 18 microorganism targets that cause respiratory symptoms, but the method involves dividing the pathogens into multiple groups [19]. Commercial kits available on the market also segment primers into groups, enabling diagnosis in a compartmentalized manner. Recently, it was developed a platform based on a multiplex recombinase polymerase amplification-assisted CRISPR-Cas12a system [20]. While promising, this approach cannot detect 22 pathogens simultaneously and requires additional reagents and time. In contrast, our methodology enables the simultaneous detection of 22 targets in a single-tube multiplex reaction with short turnaround time within 4 h, with results analyzed through capillary electrophoresis. This streamlined approach provides a more efficient and cost-effective method for diagnosing respiratory pathogens.

Our findings did not identify any samples with co-infection; however, the results indicate that the assay effectively detects multiple pathogens by reliably distinguishing between those with amplicons of similar sizes.

The use of multiplex assays capable of simultaneously identifying various viruses and bacteria responsible for flu-like symptoms and respiratory syndromes represents a significant advancement in clinical practice and hospital management. These tests provide rapid and accurate diagnoses, enabling more targeted therapeutic interventions, such as the appropriate prescription of antivirals and the reduction of indiscriminate antibiotic use. Moreover, they contribute to shorter hospital stays and reduced healthcare costs while enhancing infection control measures through the early isolation of infected patients. Currently, there is no existing multiplex diagnostic test that can detect all 22 targets like the one in this study. The tests used today for each infectious agent are considered the gold standard on their own. However, our goal is to develop a single, rapid, and easy-to-collect test that can simultaneously identify all 22 infectious agents. Because we collected a limited number of samples and only identified five of the infectious agents, we were only able to calculate the receiver operating characteristic (ROC) curve for COVID-19. For COVID-19, the test showed a sensitivity of 94.23% and a specificity of 100%. Despite this, our positive controls present in the designed plasmids demonstrate that the multiplex test has the potential to identify all 22 pathogens. Further studies with a larger number of patients will be needed to accurately determine the sensitivity and specificity of this proposed test. Thus, the adoption of such tests improves clinical safety and efficiency in the management of acute respiratory infections, with a direct impact on public health and patient outcomes. Additionally, the identification of the 22 pathogens can be used as epidemiological tools, in surveillance, and in preparedness for epidemics/pandemics.

## 5. Conclusions

Since the onset of the COVID-19 pandemic, there has been a growing demand for rapid and comprehensive diagnostic tests for respiratory diseases. This work presents an economical and simple method that enables the simultaneous detection of 22 targets. The designed primers can amplify specific regions of each target, and the grouping strategy according to the fluorophores makes it possible to discriminate between targets with similar sizes. This approach provides patients with faster diagnoses—within 4 h—leading to more appropriate treatments and better outcomes. Overall, it represents a valuable tool in the fight against severe acute respiratory infections.

## 6. Patents

This work is protected by a patent registered with the Instituto Nacional de Propriedade Industrial (INPI) under the number BR 10 2023 008493 1.

## Figures and Tables

**Figure 1 idr-17-00093-f001:**
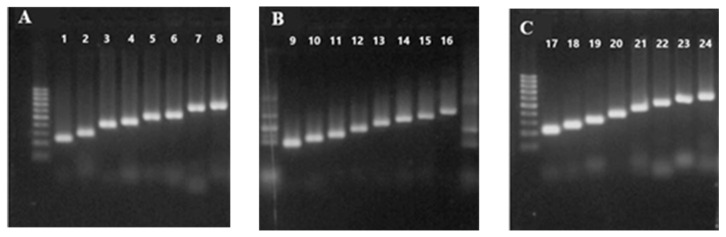
Electrophoresis in a 3% agarose gel was performed on the singleplex PCR products for all 24 targets, using the engineered plasmid. The panel was grouped according to the fluorophore and the size of the amplified fragments: (**A**) PUC18PDR-Vic containing HPeV (116 pb), RSV-A (148 pb), HPIV-1 (387 pb), HPIV-3 (200 pb), HMPV-B (264 pb) and Mpneu (292 pb). (**B**) PUC18 PDR-Fam containing EV (166 pb), IAV (127 pb), HPIV-4 (156 pb), Cor-63 (161 pb), HPIV-2 (218 pb), IBV (249 pb), Cor-229 (283 pb) and HKU (334 pb). (**C**) PUCPDR-Ned HMPVA (107 pb), Cor-43 (137 pb), AdV (166 pb), H1N1 (199 pb), RV (239 pb), HBoV (286 pb), SARS-CoV-2N2 (310 pb), RSV-B (317 pb).

**Figure 2 idr-17-00093-f002:**
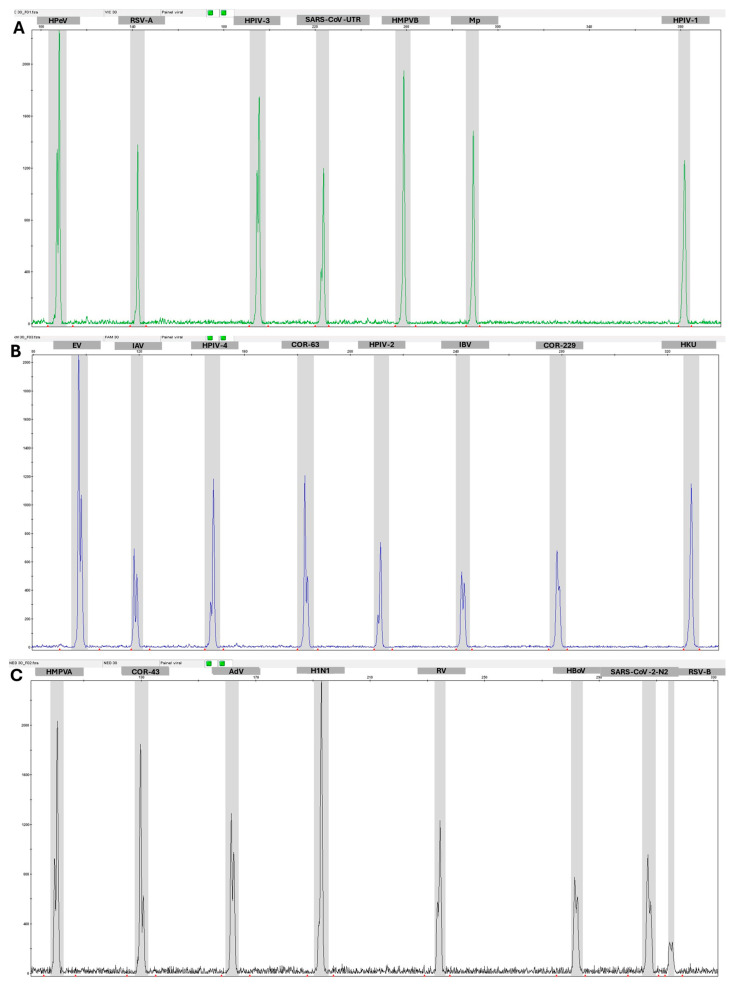
Peaks obtained in capillary electrophoresis of PCR reactions using plasmids: (**A**) PUC18PDR-Vic containing HPeV (116 pb), RSV-A (148 pb), HPIV-1 (387 pb), HPIV-3 (200 pb), SARS-CoV-2 PC (228 PB), HMPV-B (264 pb) and MPneu (292 pb). (**B**) PUC18 PDR-Fam containing EV (166 pb), IAV (127 pb), HPIV-4 (156 pb), Cor-63 (161 pb), HPIV-2 (218 pb), IBV (249 pb), Cor-229 (283 pb) and HKU (334 pb). (**C**) PUCPDR-Ned containing HMPVA (107 pb), Cor-43 (137 pb), AdV (166 pb), H1N1 (199 pb), RV (239 pb), HBoV (286 pb), SARS-CoV-2N2 (310 pb) and RSV-B (317 pb).

**Figure 3 idr-17-00093-f003:**
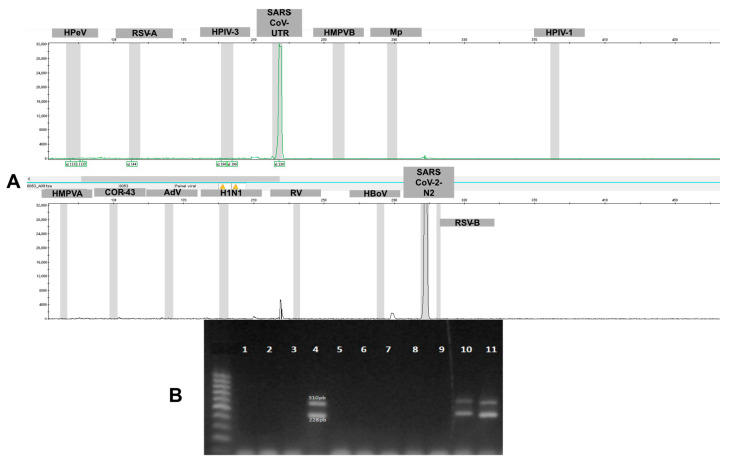
The figure shows a representative result from 11 samples from patients with flu-like symptoms and three SARS-CoV-2 positive sample amplified using the oligonucleotides described in this study. (**A**) Two SARS-CoV-2 genomic regions were targeted for validation: the 5′UTR region (228 bp) and the N2 gene region (310 bp). The capillary electrophoresis electropherogram displays two distinct peaks corresponding to the expected fragment sizes. (**B**) The 3% agarose gel confirms amplification through the presence of two bands at the expected positions in samples 4, 10 e 11. Lanes that do not show visible bands correspond to negative samples. A 1 kb DNA ladder (Ludwig Biotec) was used as the molecular weight marker.

**Figure 4 idr-17-00093-f004:**
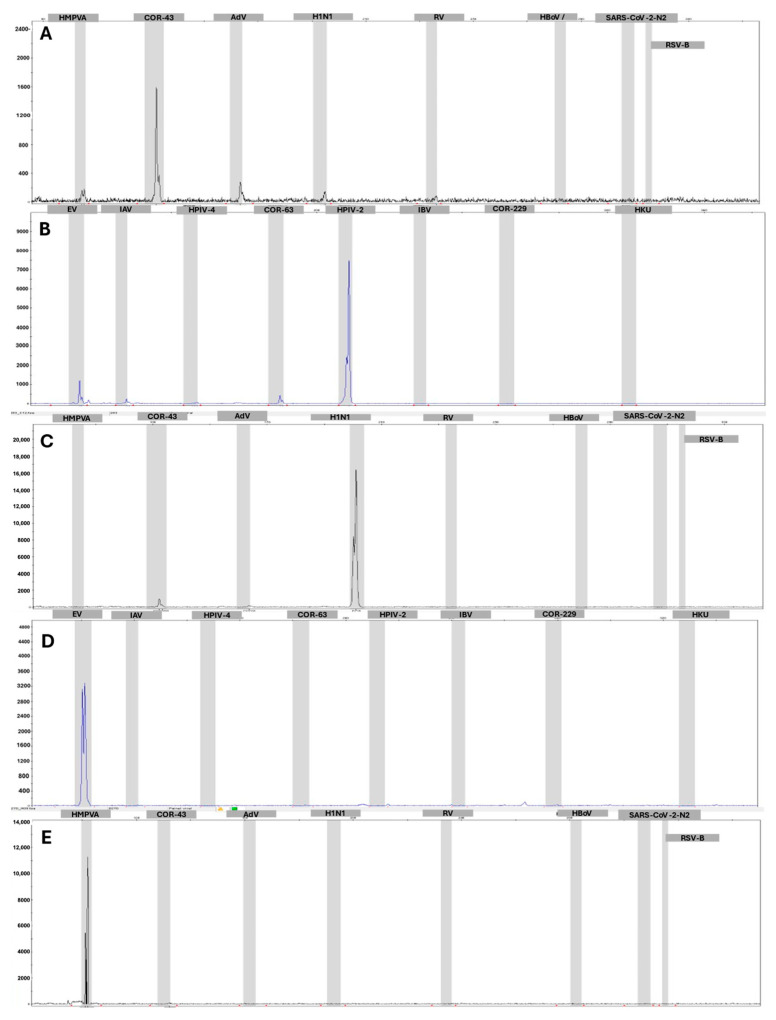
Representative electropherograms of positive samples for different respiratory viruses detected by the multiplex MPCR-CE method. (**A**) HCoV-OC43 (NED-labeled primer); (**B**) HPIV-2 (FAM); (**C**) Influenza B virus (H1N1) (NED); (**D**) Enterovirus (FAM); and (**E**) HMPV-A (NED). Each panel displays the set of targets labeled with the same fluorophore. All samples underwent a single-tube reaction containing all oligonucleotides.

**Table 1 idr-17-00093-t001:** Pathogens and their respective amplicons, oligonucleotide sequences and GenBank identification.

Target	Amplicon (pb)	Sequence Foward/Reverse, and Associated Fluorophores	GenBank Identification
**enterovirus** **(EV)**	166	CCCTGAATGCGGCTAATCC Fam-CACGGACACCCAAAGTAGTCG	HQ456309.1
**influenza A virus** **(IAV)**	127	Fam-TGAAGTTGGCAACAGGAATGC TGATGCCTGAARCCRTACC	MK902667.1
**human parainfluenza virus 4 (HPIV-4)**	156	Fam-CCTGGRGTCCCATCMAAAGTAAGT GGTTCCAGAYAAWATGGGTCTTGC	MN369047.1
**HCoV-NL63 (Cor-63)**	191	TGTTAATACACGCAATGCCACTG Fam-CATGCTTAGAGCCCAACACCA	AY567487.2
**human parainfluenza virus 2 (HPIV-2)**	218	Fam-TGGGACGCCTAAATATGGACC AGATTGGAAATGCYGCAGC	AF533012.1
**influenza B virus** **(IVB)**	249	Fam-TCGCTGTTTGGAGACACAATTG TGACAGGGGCTCTGTGATGA	CY173994.1
**HCoV-229E** **(Cor-229)**	283	TGACACYTGGGCWAAYTGGG Fam-ACCTGAAGCCAATCTATGTCCG	KU291448.1
**HCoV-HKU1** **(HKU)**	334	Fam-CTTCTTGGGCTGACCAATCTG GATGCATTGGCATATGGGC	NC_006577.2
**human parechovirus** **(HPeV)**	116	AGCCAAGGTTTARCAGACCCTTTA Vic-CATCCTTCGTGGGCCTTACA	JX575746.1
**respiratory syncytial virus A (RSV-A)**	148	GCAAATCAATGTCACTAACACCATTAG Vic-GGTCTCATGTCTGTGATCATCAGTC	MT421273.1
**human parainfluenza virus 3 (HPIV-3)**	200	Vic-TCAGCCGGTGGAGCTATCAT AGCTCTGGATTGGCATAAGCC	NC_001796.2
**human metapneumovirus B** **(HMPV-B)**	264	Vic-AAGCAGCGAACAGACAACCTG ATTCCTTAAATAATGGTGGCGC	KU320966.1
** *mycoplasma* ** ** *pneumoniae* ** **(** **Mpneu)**	292	Vic-GACACTTCACAAGTACCACCACG CGTAACGCAAAGGTGGTTGAT	NC_000912.1
**human parainfluenza virus 1 (HPIV-1)**	387	Vic-AGGGTTAAAGACAATCCAGCCA GGATCCCGCTTTGTACTGAACT	U70948.1
**human metapneumovírus A (HMPVA)**	107	AGCAGCACAGGAGAAAGACCA Ned-CTTGCAGATGCCTGTGGGT	MK357775.1
**HCoV-OC43** **(Cor-43)**	137	Ned-CTATCTGGGAACAGGACCGC TAGCCTCATCGCTACTTGGGTC	KU131570.1
**adenovirus** **(AdV)**	166	CCTTGCTACCAAAGACCGCT Ned-CCCAGTCAGCAACTTCATGGT	MT505272.1
**influenza B virus** **(H1N1)**	199	Ned-ACAACCGCAAATGCAGACAC GGATCCAGCCAGCAATGTTAC	MT505272.1
**human rhinovirus** **(RV)**	239	Ned-TCATCAGCTGGTCAATCACTGTC ACGTCACTAGCATCAGTATCTTCCA	KY093627.1
**human bocavirus** **(HBoV)**	286	Ned-CATAAACACGCCCAGGAAGTG CGTTCAGTCCCAGGAGCAAG	MG195156.1
**SARS-CoV-2N2** **(SARS-CoV-2N2)**	310	CTGATTACAAACATTGGCCGC Ned-TCTGCAGCAGGAAGAAGAGTCAC	MT396241.1
**respiratory syncytial virus B (RSV-B)**	317	CCGCCAATCCAATACATACATTG Ned-TGTGATGCCATGACTCTGTGAG	MH327947.1

**Table 2 idr-17-00093-t002:** Respiratory disease panel grouping of targets based on different fluorophores.

FAM	VIC	NED
EV	HPeV	HMPVA
IAV	RSV-A	Cor-43
HPIV-4	HPIV-3	AdV
Cor-63	SARS-CoV-2 UTR	H1N1
HPIV-2	HMPV-B	RV
IVB	Mpneu	HBoV
Cor-229	HPIV-1	SARS-CoV-2N2
HKU		RSV-B

## Data Availability

Data is available upon reasonable request to the corresponding author.

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
