# Peer review of "Detection of Microorganisms Causing Human Respiratory Infection Using One-Tube Multiplex PCR"

_2036-7449, 2025, doi:10.3390/idr17040093_

Round 1
Reviewer 1 Report
Comments and Suggestions for Authors
The laboratory diagnostic is essential step of each therapeutic intervention, but also is important from epidemiological tool, such as surveillance, and epidemic/pandemic preparedness. Thanks to the COVID-19 pandemic, the molecular biology techniques, such as PCR, RT-PCR, RT-qPCR, LAMP, are more common in the diagnostic laboratories, and thus the gold standards for detecting of infectious agent of many diseases with molecular biology techniques are so-called 'gold standards'. Lima and colleagues, in their manuscript presented a new detection technique of 22 respiratory pathogens using qRT-PCR and capillary electrophoresis. The article is interesting and shows the strength of molecular techniques in the field of medicine. However, I found several flaws, which you can find below:
MAJOR COMMENTS:
- While the idea of RT-qPCR-CE is interesting, the reality of use of such equipment is very low. Probably, only few hospitals in the European Union have Applied Biosciences 3500 sequencer. Such equipment are usually in the hub facilities at the universities. So can you calculate what is the real costs of your test.
- Authors presented a novel diagnostic technique, but did not explain why anyone should use it. Except of influenza A virus and SARS-CoV-2, there are no FDA/EMA-approved therapeutic, so why identify infectious agent?
- Please describe in details the sample collection procedure. Please indicate the inclusion and exclusion criteria. Please also characterize the group, including information whether the group was representative.
- When you propose novel diagnostic method, you must follow the rules of the old ones and use their parameters. So what is your specificity, sensitivity and limit of detection? And please compare these parameters with other molecular biology diagnostic methods.
- The Figures 2, 3, 4 are not readable. Please replace.
- Please rephrase the explanation of no-positive tests for influenza. It would be the best to compare it with the official data of confirmed cases of flu and COVID-19 within the period of sample collection from the same region/city.
- I am also surprised that authors did not detect any co-infections, as well as did not validate their method for such occasion. It would be very helpful to see such analysis.
MINOR COMMENTS:
1. Please use correct names for each detected virus, with appropriate abbreviations, such as influenza A virus (IAV), respiratory syncytial virus (RSV), human metapneumovirus (HMPV), etc.
Comments on the Quality of English LanguageI do not qualify to judge the quality of English language.
Author Response
|
Comments 1: While the idea of RT-qPCR-CE is interesting, the reality of use of such equipment is very low. Probably, only few hospitals in the European Union have Applied Biosciences 3500 sequencer. Such equipment are usually in the hub facilities at the universities. So can you calculate what is the real costs of your test.
|
|
Response 1: Indeed, it is a high-cost instrument, and not all facilities have access to it or the means to acquire one. However, the low final cost per reaction makes it feasible to outsource the test to centers equipped with the sequencer, resulting in a commercially viable solution for both the performing and the contracting institutions. Currently, we estimate that the consumables required for the sequencer— including plastics, polymer, capillary, wen, formamide, primers, conditioning reagent, anode buffer, and cathode buffer — amount to approximately US$6.00 (six dollars) per sample.
|
|
Comments 2: Authors presented a novel diagnostic technique, but did not explain why anyone should use it. Except of influenza A virus and SARS-CoV-2, there are no FDA/EMA-approved therapeutic, so why identify infectious agent?
|
|
Response 2: The use of multiplex assays capable of simultaneously identifying various viruses and bacteria responsible for flu-like symptoms and respiratory syndromes represents a significant advancement in clinical practice and hospital management. These tests provide rapid and accurate diagnoses, enabling more targeted therapeutic interventions, such as the appropriate prescription of antivirals and the reduction of indiscriminate antibiotic use. Moreover, they contribute to shorter hospital stays and reduced healthcare costs, while enhancing infection control measures through the early isolation of infected patients. Their high sensitivity (94,23%) and specificity (100%) further underscore their role as a strategic tool for epidemiological surveillance and the detection of coinfections. Thus, the adoption of such tests improves clinical safety and efficiency in the management of acute respiratory infections, with a direct impact on public health and patient outcomes. Additionally, the identification of the 22 pathogens can be used as epidemiological tools, in surveillance, and in preparedness for epidemics/pandemics.
Comments 3: Please describe in details the sample collection procedure. Please indicate the inclusion and exclusion criteria. Please also characterize the group, including information whether the group was representative. Response 3: Samples were obtained through a partnership with a private clinical laboratory. All patients who requested molecular testing for COVID-19 were invited to participate in the study. Those who agreed followed the laboratory's standard collection procedures for molecular detection of SARS-CoV-2, which included nasopharyngeal swabs or the collection of 3 to 5 mL of saliva. The samples type and collection methods were already standardized for SARS-CoV-2 molecular testing. Samples were excluded if they were received with a dry swab or with less than 3 mL of saliva. Due to the context of the pandemic, the study population was characterized by the presence of various respiratory symptoms. Any patient presenting one or more respiratory symptoms resembling flu-like or cold-like conditions, or who had contact with a confirmed COVID-19 case, was included in the study group. This information was added in the section Materials and Methods.
Comments 4: When you propose novel diagnostic method, you must follow the rules of the old ones and use their parameters. So what is your specificity, sensitivity and limit of detection? And please compare these parameters with other molecular biology diagnostic methods. Response 4: Currently, there is no existing multiplex diagnostic test that can detect all 22 targets like the one in this study. The tests used today for each infectious agent are considered the gold standard on their own. However, our goal is to develop a single, rapid, and easy-to-collect test that can simultaneously identify all 22 infectious agents. Because we collected a limited number of samples and only identified five of the infectious agents, we were only able to calculate the Receiver Operating Characteristic (ROC) curve for COVID-19. For COVID-19, the test showed a sensitivity of 94.23% and a specificity of 100%. Despite this, our positive controls present in the designed plasmids demonstrated that the multiplex test has the potential to identify all 22 pathogens. Further studies with a larger number of patients will be needed to accurately determine the sensitivity and specificity of this proposed test.
Comments 5: The Figures 2, 3, 4 are not readable. Please replace. Response 5: The figures were replaced.
Comments 6: Please rephrase the explanation of no-positive tests for influenza. It would be the best to compare it with the official data of confirmed cases of flu and COVID-19 within the period of sample collection from the same region/city. Response 6: Thank you for your suggestion, the phrase was rewritten: These factors might explain why we didn't detect the influenza virus in our samples. Additionally, the samples were collected during the COVID-19 pandemic, which could have influenced the results (pag. 12).
Comments 7: I am also surprised that authors did not detect any co-infections, as well as did not validate their method for such occasion. It would be very helpful to see such analysis. Response 7: We also understand that these analyses can identify co-infection, but in the samples analyzed in this work, no co-infection was found, as shown in the figures 2, 3 and 4. MINOR COMMENTS: 1. Please use correct names for each detected virus, with appropriate abbreviations, such as influenza A virus (IAV), respiratory syncytial virus (RSV), human metapneumovirus (HMPV), etc.
Response: All the correct names with their appropriate abbreviations have been included and are highlighted in red.
|
|
4. Response to Comments on the Quality of English Language |
|
Point 1: |
|
Response 1: We have revised the English version to make it clearer and more fluent.
|
|
5. Additional clarifications |
|
We truly appreciate the feedback from the reviewers. We thank them for the comprehensive reading of this manuscript and their comments and suggestions, which have helped us to improve the quality of our text and broaden our knowledge. Please, find the answers to the points raised on the following pages and the Introduction, Materials and Methods, Results, Discussion and Conclusion have been improved like reviewer suggestions. |
Reviewer 2 Report
Comments and Suggestions for Authors
There is a growing demand for multiplex PCR technology, and further advancements in this field are anticipated. This study merits publication in this journal, as it successfully demonstrated the simultaneous detection of 20 different pathogens using a multiplex PCR approach. However, merely reporting successful primer combinations and reaction conditions is insufficient for guiding researchers aiming to develop large-scale multiplex detection systems in the future.
Multiplex PCR research has three principal scientific contributions:
1. **Provision of validated primer sets**: Offering researchers access to primer sequences and their combinations that have been experimentally confirmed to work.
2. **Dissemination of design strategies**: Sharing detailed methodologies and design rationales that enable other researchers to create effective primer sets for future multiplex detection, thus contributing to the broader development of multiplex PCR technologies.
3. **Methodological enhancement through advanced technologies**: Introducing future-oriented tools, such as AI-based primer design, to improve the accuracy and efficiency of multiplex primer development.
Given the current sophistication of PCR design technologies, and the increasing need to optimize target selection and detection sensitivity, it is essential that the authors go beyond reporting success. They should also include primer design criteria and examples of unsuccessful primer sets in the supplementary materials, to inform and improve future research.
### Comments and Recommendations:
* **Regarding Point 1**: The manuscript does not include a rationale for pathogen selection. Please provide specific selection criteria or indicators, particularly for respiratory pathogens detected in non-invasive samples such as saliva.
* **Regarding Point 2**: The explanation of the primer design process is insufficient. Please address the following aspects with detailed, reproducible data:
* **2.1 Melting Temperature (Tm) Range**: Since all primers in multiplex PCR must work under the same thermal cycling conditions, provide the predicted Tm values and the method used for their calculation. Typically, primers are designed within a Tm range of 56°C to 62°C. As the cycling conditions in this study included a 59°C annealing temperature, it is assumed that the Tm values fall within or near this range—this must be explicitly stated and justified.
* **2.2 Annealing Temperature Optimization**: Include in the Supplement the range of annealing temperatures tested and describe the process used to determine the optimal annealing temperature.
* **2.3 Primer Complementarity Check**: Clarify how primer-dimer formation was evaluated, especially at the 3' end. For instance, it is generally recommended to avoid perfect complementarity among the last five bases at the 3' end. Describe the computational method used, including the thermodynamic model (e.g., ΔG thresholds) applied for dimer prediction.
* **2.4 Secondary Structure Avoidance**: Similarly, provide the method and ΔG thresholds used to evaluate and avoid secondary structures within individual primers.
* **Regarding Point 3**: AI-based approaches to primer design are expected to play a key role in the future development of multiplex PCR. Please cite:
*Kayama, Kotetsu, et al. "Prediction of PCR amplification from primer and template sequences using recurrent neural network." Scientific Reports 11.1 (2021): 7493.*
This work supports the importance of accumulating design-performance data for machine learning applications in primer design.
* **Additional Request**: Since this study involved a large number of primers in a single reaction, please disclose in the supplementary materials the primer combinations that were not successfully amplified when tested together. This negative data is crucial for future research to identify incompatible primer pairings in multiplex settings.
---
**Conclusion**:
With the inclusion of detailed primer design parameters, pathogen selection criteria, unsuccessful primer combinations, and forward-looking AI integration, this manuscript can provide a highly valuable contribution to the field of multiplex PCR.
**With the advances in AI, it is now possible to create an AI that predicts the success or failure of PCR as long as primers and PCR results are accumulated, and it is possible to improve the accuracy and sensitivity of multiplex PCR as described in the submitted paper. If the authors mention this and publish the negative data necessary for future consideration in the Supplement, I believe that there is a high possibility that researchers who perform PCR prediction analysis, which is expected to increase in the future, will refer to your journal.
Author Response
|
Comments 1: Multiplex PCR research has three principal scientific contributions: 1. **Provision of validated primer sets**: Offering researchers access to primer sequences and their combinations that have been experimentally confirmed to work.
|
|
Response 1: The primer sequences are listed in Table 1. They work in separate reactions as well as together. This can be seen in Figure 1, where all primers were tested with the plasmids, and in Figure 4, where—despite being displayed in separate electropherograms—the reaction was performed with all primers in a single reaction tube. The separation is achieved through the filter that detects each fluorophore (VIC, FAM, and NED).
|
|
Comments 2: **Dissemination of design strategies**: Sharing detailed methodologies and design rationales that enable other researchers to create effective primer sets for future multiplex detection, thus contributing to the broader development of multiplex PCR technologies.
|
|
Response 2: The 'Targets and Primer Design' section has been rewritten to provide a clearer explanation of the primer design process. The following text was included in the Materials and Methods section, subsection 2.2 (page 03): The diagnostic platform was developed to detect a broad panel of respiratory pathogens, including adenovirus, enterovirus, influenza A (with H1N1 subtype), influenza B, human parainfluenza viruses 1–4, four human coronaviruses (NL63, 229E, HKU1, and OC43), human parechovirus, respiratory syncytial viruses A and B, human metapneumoviruses A and B, Mycoplasma pneumoniae, human rhinovirus, human bocavirus, and SARS-CoV-2. Target sequences were retrieved from the NCBI GenBank database and aligned using Clustal Omega [4] to identify regions of high sequence similarity across strains, which were then prioritized for primer design to ensure broad detection capability. These conserved regions were selected to maximize the primers' ability to recognize diverse genetic variants of each pathogen. Oligonucleotides were designed using Primer Express® 3.0.1 (ThermoFisher Scientific), with each primer pair tailored to amplify sequences of approximately 20 base pairs, featuring an annealing temperature near 60°C and a GC content between 40–60% to balance stability and specificity. Following initial design, the primers were analyzed in Primer Explorer to evaluate potential secondary structures, including hairpins and primer-dimers, and to confirm optimal annealing temperatures. This step was critical to minimize off-target interactions and ensure robust amplification efficiency. To address overlapping amplicon sizes, a multiplexing strategy was implemented, organizing the primers into three groups, each labeled with a distinct fluorophore (VIC, FAM, or NED). Within each group, amplicon sizes were designed to differ by at least 20 base pairs, enabling clear discrimination during capillary electrophoresis. By leveraging conserved genomic regions for primer design and combining fluorophore labeling with precise amplicon sizing, the platform achieved simultaneous detection and differentiation of all 22 targets in a single, high-throughput assay.
Comments 3: **Methodological enhancement through advanced technologies**: Introducing future-oriented tools, such as AI-based primer design, to improve the accuracy and efficiency of multiplex primer development. Given the current sophistication of PCR design technologies, and the increasing need to optimize target selection and detection sensitivity, it is essential that the authors go beyond reporting success. They should also include primer design criteria and examples of unsuccessful primer sets in the supplementary materials, to inform and improve future research. Response 3: We agree with the reviewer, but in this study the primers were not created using these tools, and we recommend that the test utilize the primers designed by our team.
### Comments and Recommendations: * **Regarding Point 1**: The manuscript does not include a rationale for pathogen selection. Please provide specific selection criteria or indicators, particularly for respiratory pathogens detected in non-invasive samples such as saliva.
Response: The pathogens were chosen based on the molecular panels that became available on the market in response to the COVID-19 pandemic, such as the BioFire FilmArray Respiratory Panel, Seegene Allplex Respiratory Panel, and Luminex NxTAG Respiratory Pathogen Panel. Our main goal was to find a more practical and cost-effective alternative to these commercial products, which are often quite expensive. Traditional molecular diagnostic methods usually require running multiple separate reactions to detect all the targets, meaning you need to buy several different kits. We aimed to develop a new diagnostic platform that is simpler to use, highly sensitive, and specific capable of detecting any target, or even multiple targets, in just a single reaction tube. We achieved this by combining differences in the sizes of the amplified DNA fragments with the use of different fluorescent labels. In other words, fragments of similar size were tagged with different fluorophores, allowing us to distinguish multiple targets simultaneously in one test.
Comments: * **Regarding Point 2**: The explanation of the primer design process is insufficient. Please address the following aspects with detailed, reproducible data: * **2.1 Melting Temperature (Tm) Range**: Since all primers in multiplex PCR must work under the same thermal cycling conditions, provide the predicted Tm values and the method used for their calculation. Typically, primers are designed within a Tm range of 56°C to 62°C. As the cycling conditions in this study included a 59°C annealing temperature, it is assumed that the Tm values fall within or near this range—this must be explicitly stated and justified.
* **2.2 Annealing Temperature Optimization**: Include in the Supplement the range of annealing temperatures tested and describe the process used to determine the optimal annealing temperature.
* **2.3 Primer Complementarity Check**: Clarify how primer-dimer formation was evaluated, especially at the 3' end. For instance, it is generally recommended to avoid perfect complementarity among the last five bases at the 3' end. Describe the computational method used, including the thermodynamic model (e.g., ΔG thresholds) applied for dimer prediction.
* **2.4 Secondary Structure Avoidance**: Similarly, provide the method and ΔG thresholds used to evaluate and avoid secondary structures within individual primers. Response: Response 2.1, 2.2, 2.3 and 2.4: The 'Targets and Primer Design' section has been rewritten to provide a clearer explanation of the primer design process. The following text was included in the Materials and Methods section, subsection 2.2 (page 03): The diagnostic platform was developed to detect a broad panel of respiratory pathogens, including adenovirus, enterovirus, influenza A (with H1N1 subtype), influenza B, human parainfluenza viruses 1–4, four human coronaviruses (NL63, 229E, HKU1, and OC43), human parechovirus, respiratory syncytial viruses A and B, human metapneumoviruses A and B, Mycoplasma pneumoniae, human rhinovirus, human bocavirus, and SARS-CoV-2. Target sequences were retrieved from the NCBI GenBank database and aligned using Clustal Omega [4] to identify regions of high sequence similarity across strains, which were then prioritized for primer design to ensure broad detection capability. These conserved regions were selected to maximize the primers' ability to recognize diverse genetic variants of each pathogen. Oligonucleotides were designed using Primer Express® 3.0.1 (ThermoFisher Scientific), with each primer pair tailored to amplify sequences of approximately 20 base pairs, featuring an annealing temperature near 60°C and a GC content between 40–60% to balance stability and specificity. Following initial design, the primers were analyzed in Primer Explorer to evaluate potential secondary structures, including hairpins and primer-dimers, and to confirm optimal annealing temperatures. This step was critical to minimize off-target interactions and ensure robust amplification efficiency. To address overlapping amplicon sizes, a multiplexing strategy was implemented, organizing the primers into three groups, each labeled with a distinct fluorophore (VIC, FAM, or NED). Within each group, amplicon sizes were designed to differ by at least 20 base pairs, enabling clear discrimination during capillary electrophoresis. By leveraging conserved genomic regions for primer design and combining fluorophore labeling with precise amplicon sizing, the platform achieved simultaneous detection and differentiation of all 22 targets in a single, high-throughput assay.
Comments: * **Regarding Point 3**: AI-based approaches to primer design are expected to play a key role in the future development of multiplex PCR. Please cite: *Kayama, Kotetsu, et al. "Prediction of PCR amplification from primer and template sequences using recurrent neural network." Scientific Reports 11.1 (2021): 7493.* This work supports the importance of accumulating design-performance data for machine learning applications in primer design. Response: We agree with the reviewer on the importance of gathering design and performance data for machine learning. We included this work in the section Discussion (pag 12)
* **Additional Request**: Since this study involved a large number of primers in a single reaction, please disclose in the supplementary materials the primer combinations that were not successfully amplified when tested together. This negative data is crucial for future research to identify incompatible primer pairings in multiplex settings.
Response: All primers worked together as a combination, as shown in Figures 1 and 2. In Figure 1, we present agarose gels separately to make it easier to see the results. Because of the limitations of this technique, bands with similar sizes (measured in base pairs, or bp) can be hard to distinguish clearly. In Figure 2, we show electropherograms based on the different fluorescent labels used for the primers. The primers were divided into three groups to prevent primers that amplify fragments of similar sizes from being labeled with the same fluorophore. To clarify further: primers targeting Enterovirus, Parechovirus, and Human Metapneumovirus produce fragments of 107 bp, 116 bp, and 107 bp, respectively. To tell these targets apart, each was labeled with a different fluorescent dye—FAM, VIC, and NED.
|
|
4. Response to Comments on the Quality of English Language |
|
Point 1: |
|
Response 1: We have revised the English version to make it clearer and more fluent.
|
|
5. Additional clarifications |
|
We truly appreciate the feedback from the reviewers. We thank them for the comprehensive reading of this manuscript and their comments and suggestions, which have helped us to improve the quality of our text and broaden our knowledge. Please, find the answers to the points raised on the following pages and the Introduction, Materials and Methods, Results, Discussion and Conclusion have been improved like reviewer suggestions.
|

Reviewer 3 Report
Comments and Suggestions for Authors
Lima and colleagues, in the manuscript titled “Detection of microorganisms causing respiratory syndrome using one-tube multiplex PCR”, describe an in-house developed multiplex platform based on PCR method for the detection of 22 respiratory pathogens. The detection is based on PCR-CE with fluorophore usage and a group of 340 samples of saliva and nasopharynx.
Respiratory pathogens, particularly viruses such as influenza, RSV, and coronaviruses, remain a major cause of global morbidity and mortality. They contribute significantly to hospitalizations, especially among vulnerable populations like children, the elderly, and immunocompromised individuals. Their rapid transmission, potential for mutation, and seasonal variability make them a constant public health challenge. Moreover, the emergence of novel respiratory viruses, such as SARS-CoV-2, underscores the need for vigilance in monitoring respiratory infections. In this context, the development of accurate, rapid, and sensitive detection methods is crucial. Traditional diagnostics often face limitations in speed, sensitivity, or scalability. Therefore, the continuous innovation and implementation of novel detection technologies—ranging from molecular diagnostics to point-of-care platforms—are not only welcome but essential. Hence, the theme of the current research is highly important.
The writing is clear and concise, demonstrating accurate and effective use of the English language. The author’s message comes through clearly, even though there are occasional issues with grammar that could benefit from refinement.
Major Comments:
- The title uses the syntagm “respiratory syndrome”. I would suggest that this be changed to “respiratory infection”, or a similar recognizable idiom, as “respiratory syndrome” is not a widely recognizable medical term, or is used in conjunction with “severe acute respiratory syndrome”. Especially since, in the Introduction portion of the Manuscript, the Authors in the last paragraph use the term “flu-like syndromes”.
- The Abstract is understandable, but can be improved. How were the primers and probes designed (GenBank), via which software (Primer Express® 3.0)? Sample types should be noted in the Methods, not the Results. Where were the samples taken from (Institution, clinical setting, volunteers…?), and in what period? Regarding the results, were the majority of positive samples from saliva, or nasopharynx swabs? The text is good, but should be written precisely and in more detail. I would kindly request that the Authors restructure their Abstract.
- In the Materials and Methods section, it would be informative to see how many samples were taken each – how many nasopharyngeal swabs and how many saliva samples. Can the Authors access information on when the samples were taken, in what period (due to the periodic patterns observed in the prevalence of certain viruses within the population)? This may influence the percentage of positive samples. Also, as stated above, what is the origin of samples (Institution, clinical setting, volunteers…?)? The origin of samples is, to my knowledge, only noted in the Acknowledgement section: “…company BioGenetics Biologia Molecular Ltda which assisted us in obtaining patient samples”?
- The Authors say that the method “facilitates faster diagnosis”, but they provide us with the turnaround time only in the Conclusions section. Since this is very important, it should be mentioned in the Results and Discussion parts of the text, not only Conclusions.
- Please consider refining the English language throughout the text. Although the text is clear and understandable, some grammatical errors may be found.
Minor comments:
- If the patient samples were used for diagnosis, which method was used for initial diagnosis? Were any commercial assays used for comparison with this method? If yes, how did their performance compare to that of the proposed method? How does this method compare to the newly devised platform of the Authors’?
- I kindly invite the Authors to use correct orthography while writing names of pathogens (e.g. Mycoplasma pneumoniae should be written in Italics; human bocavirus should be written in small letters, when writing the virus name).
- Please define the abbreviations “CE” and “RT” in the Abstract, as that is their first mention.
- If the term “qPCR” is in the Keywords of the Abstract, please place the term in the Abstract text as well.
- The subtitles in the 2. Materials and Methods section should be in italics, and numbered 2.1, 2.2…, etc.
- Names of manufacturers, their city and country should be noted with each commercial reagent or apparatus used (e.g. this information is missing next to Applied Biosciences 3500 ® in line 96).
Summary: After reviewing the present study, in my opinion it may be considered to fall in the “Minor revisions” category. After addressing my concerns regarding the Abstract, Title and Methods, I would recommend the work for publication. To conclude, the Authors must respond to the Major comments, while their response to the Minor comments is left to their discretion.
I would like to express my gratitude to the Authors for taking the time to consider my comments and suggestions.
Author Response
|
Comments 1: The title uses the syntagm “respiratory syndrome”. I would suggest that this be changed to “respiratory infection”, or a similar recognizable idiom, as “respiratory syndrome” is not a widely recognizable medical term, or is used in conjunction with “severe acute respiratory syndrome”. Especially since, in the Introduction portion of the Manuscript, the Authors in the last paragraph use the term “flu-like syndromes”.
|
|
Response 1: We appreciate the feedback and have changed the title and the term “flu-like syndromes”.
|
|
Comments 2: The Abstract is understandable, but can be improved. How were the primers and probes designed (GenBank), via which software (Primer Express® 3.0)? Sample types should be noted in the Methods, not the Results. Where were the samples taken from (Institution, clinical setting, volunteers…?), and in what period? Regarding the results, were the majority of positive samples from saliva, or nasopharynx swabs? The text is good, but should be written precisely and in more detail. I would kindly request that the Authors restructure their Abstract.
|
|
Response 2: Thank you for your suggestion. I have rewritten the abstract, and the changes are highlighted in red.
Comments 3: In the Materials and Methods section, it would be informative to see how many samples were taken each – how many nasopharyngeal swabs and how many saliva samples. Can the Authors access information on when the samples were taken, in what period (due to the periodic patterns observed in the prevalence of certain viruses within the population)? This may influence the percentage of positive samples. Also, as stated above, what is the origin of samples (Institution, clinical setting, volunteers…?)? The origin of samples is, to my knowledge, only noted in the Acknowledgement section: “…company BioGenetics Biologia Molecular Ltda which assisted us in obtaining patient samples”?
Response 3: Samples were collected in collaboration with a private laboratory. All patients visiting the lab for SARS-CoV-2 testing were invited to participate in the study by providing their samples. Typically, each patient provided one sample, either saliva or a nasopharyngeal swab, resulting in 289 nasopharyngeal samples and 51 saliva samples during September 2020 to September 2021. Patients who agreed to participate signed an informed consent form. After their COVID-19 test was completed, they donated any remaining samples for research purposes.
Comments 4: The Authors say that the method “facilitates faster diagnosis”, but they provide us with the turnaround time only in the Conclusions section. Since this is very important, it should be mentioned in the Results and Discussion parts of the text, not only Conclusions. Response 4: Thank you for your suggestion. The text has the turnaround time in the Results part (pag. 11, line 237), but we included it in the Discussion part (pag. 12, line 278).
Comments 5: Please consider refining the English language throughout the text. Although the text is clear and understandable, some grammatical errors may be found. Response 5: We have revised the English version to make it clearer and more fluent.
Comments 6: If the patient samples were used for diagnosis, which method was used for initial diagnosis? Were any commercial assays used for comparison with this method? If yes, how did their performance compare to that of the proposed method? How does this method compare to the newly devised platform of the Authors’? Response 6: The samples were initially tested using the Allplex SARS-Cov2 Assay Kit for SARS-CoV-2. These results were then used for comparison and validation of the COVID-19 diagnosis obtained through our multiplex method combined with capillary electrophoresis. The other targets were compared using Allplex® qPCR kits.
Comments 7: I kindly invite the Authors to use correct orthography while writing names of pathogens (e.g. Mycoplasma pneumoniae should be written in Italics; human bocavirus should be written in small letters, when writing the virus name). Response 7: Thank you for your feedback, we changed the names of pathogens and the changes are highlighted in red.
Comments 8: Please define the abbreviations “CE” and “RT” in the Abstract, as that is their first mention.
Response 8: Thank you for your feedback, we included the definition of the abbreviations in the Abstract.
Comments 9: If the term “qPCR” is in the Keywords of the Abstract, please place the term in the Abstract text as well. Response 9: Thank you for your feedback. We changed the abstract, which no longer has qPCR, so we included RT-PCR in the keywords, which better represents the technique used in the work.
Comments 10: The subtitles in the 2. Materials and Methods section should be in italics, and numbered 2.1, 2.2…, etc. Response 10: Thank you for your feedback, we included the numbers and the text in italics.
Comments 11: Names of manufacturers, their city and country should be noted with each commercial reagent or apparatus used (e.g. this information is missing next to Applied Biosciences 3500 ® in line 96). Response 11: We included this information Applied Biosciences 3500xL ® Genetic Analyser (Paisley-Scotland) in the text (pag. 04).
|
|
4. Response to Comments on the Quality of English Language |
|
Point 1: |
|
Response 1: We have revised the English version to make it clearer and more fluent.
|
|
5. Additional clarifications |
|
We truly appreciate the feedback from the reviewers. We thank them for the comprehensive reading of this manuscript and their comments and suggestions, which have helped us to improve the quality of our text and broaden our knowledge. Please, find the answers to the points raised on the following pages and the Introduction, Materials and Methods, Results, Discussion and Conclusion have been improved like reviewer suggestions.
|

Round 2
Reviewer 1 Report
Comments and Suggestions for Authors
I am grateful to the authors for the responses to my comments regarding their manuscript on the novel method for the respiratory pathogen detection based on capillary electrophoresis. However, I am pretty convinced that similar questions/comments will arise in the heads of interested readers of Infectious Disease Reports. Thus, I strongly recommend implementing the answers to my comments into the manuscript, especially in the Discussion section. Moreover, I suggest adding to the manuscript the analysis of novel MPCR-CE detection method's sensitivity and specificity, even if it is only for SARS-CoV-2. In addition, I have several minor comments, as listed below.
- In the abstract authors wrote: 'Positive results were detected in both nasopharyngeal and saliva swabs, with SARS-CoV-2 predominating in saliva samples'. However, this observation is not described in any other manuscript section. Please include these results in the manuscript.
- I will repeat my previous comment - influenza is a disease, and the influenza A/B virus is a pathogen causing the disease. Please change the manuscript accordingly.
- In Table 1:
- 'influenza A' appears two times with different amplicons sizes
- there are no second pair of primers for SARS-CoV-2 UTR
- please unify and be consistent with the abbreviations of viruses, i.e. Hcov-Oc43 in Table, and then HCoV-OC43 in. the text. The 'HCoV-OC43' is correct!
- it would be good to change the order of the viruses and compile them in some groups, at least put the different primers for 1 virus next to each other, and group some viruses, i.e., coronaviruses
- In Table 2, SARS-CoV-2 UTR is missing in the first column as I understand correctly.
- Figure 3 is ugly and hard to read, sorry! Needs to be corrected and improved, i.e., there is no need to include the whole agarose gel with only 3 lanes loaded with samples. The legend is also quite modest, can be elucidated.
- Figure 4 - the peaks for the same virus should be at the same length in different graphs and directly underneath
Author Response
Dear Reviewer,
We truly appreciate your feedback. We thank you for the comprehensive reading of this manuscript which have helped us to improve the quality of our text and broaden our knowledge.
Please, find the answers to the points raised on the following pages.
Thus, I strongly recommend implementing the answers to my comments into the manuscript, especially in the Discussion section. Moreover, I suggest adding to the manuscript the analysis of novel MPCR-CE detection method's sensitivity and specificity, even if it is only for SARS-CoV-2.
Response: Thank you for your feedback. We added the text below in the Discussion section:
The use of multiplex assays capable of simultaneously identifying various viruses and bacteria responsible for flu-like symptoms and respiratory syndromes represents a significant advancement in clinical practice and hospital management. These tests provide rapid and accurate diagnoses, enabling more targeted therapeutic interventions, such as the appropriate prescription of antivirals and the reduction of indiscriminate antibiotic use. Moreover, they contribute to shorter hospital stays and reduced healthcare costs, while enhancing infection control measures through the early isolation of infected patients. Currently, there is no existing multiplex diagnostic test that can detect all 22 targets like the one in this study. The tests used today for each infectious agent are considered the gold standard on their own. However, our goal is to develop a single, rapid, and easy-to-collect test that can simultaneously identify all 22 infectious agents. Because we collected a limited number of samples and only identified five of the infectious agents, we were only able to calculate the Receiver Operating Characteristic (ROC) curve for COVID-19. For COVID-19, the test showed a sensitivity of 94.23% and a specificity of 100%. Despite this, our positive controls present in the designed plasmids demonstrate that the multiplex test has the potential to identify all 22 pathogens. Further studies with a larger number of patients will be needed to accurately determine the sensitivity and specificity of this proposed test. Thus, the adoption of such tests improves clinical safety and efficiency in the management of acute respiratory infections, with a direct impact on public health and patient outcomes. Additionally, the identification of the 22 pathogens can be used as epidemiological tools, in surveillance, and in preparedness for epidemics/pandemics.
In addition, I have several minor comments, as listed below.
In the abstract authors wrote: 'Positive results were detected in both nasopharyngeal and saliva swabs, with SARS-CoV-2 predominating in saliva samples'. However, this observation is not described in any other manuscript section. Please include these results in the manuscript.
Response: We included the phrase “'Positive results were detected in both nasopharyngeal and saliva swabs, with SARS-CoV-2 predominating in saliva samples” in the page 12, Results Section.
I will repeat my previous comment - influenza is a disease, and the influenza A/B virus is a pathogen causing the disease. Please change the manuscript accordingly.
Response: Thank you for your comment. We changed the manuscript accordingly.
In Table 1: 'influenza A' appears two times with different amplicons sizes
there are no second pair of primers for SARS-CoV-2 UTR
Response: Table 1 has been corrected; the virus with a 199 bp amplicon is Influenza B Virus (H1N1). The change has been marked red.
The second primer pair targeting the SARS-CoV-2 5'UTR is not part of the newly designed oligonucleotides developed specifically for this study. It is included in a separate work currently under patent application. We chose to incorporate it into this study to align with the gold standard diagnostic approach for COVID-19 at the time, which recommended targeting two genomic regions for reliable detection.
Please unify and be consistent with the abbreviations of viruses, i.e. Hcov-Oc43 in Table, and then HCoV-OC43 in. the text. The 'HCoV-OC43' is correct!
Response: Thank you, the text was corrected.
It would be good to change the order of the viruses and compile them in some groups, at least put the different primers for 1 virus next to each other, and group some viruses, i.e., coronaviruses
Response: We grouped the viruses based on the fluorophore and the size of the amplified fragments to make it easier to understand the constructed panel.
In Table 2, SARS-CoV-2 UTR is missing in the first column as I understand correctly.
Response: It was indeed missing and has now been added to Table 2.
Figure 3 is ugly and hard to read, sorry! Needs to be corrected and improved, i.e., there is no need to include the whole agarose gel with only 3 lanes loaded with samples. The legend is also quite modest, can be elucidated.
Response: Unfortunately, our equipment does not allow for higher-quality imaging.
Regarding the agarose gel, the lanes without bands correspond to samples from patients who tested negative for SARS-CoV-2. This information has been added to the manuscript and the figure legend has been revised to better clarify the data presented.
Revised section of the text: Line 228- “As illustrated in Figure 3, amplification of SARS-CoV-2 (PC) (228-bp) and SARS-CoV-2 N2 (310 bp) showed the expected peaks in CE Figure 3A, from a patient with a positive SARS-CoV-2 diagnosis. Figure 3B shows results for eleven patient samples analyzed on a 3% agarose gel and three samples showed the two expected bands for SARS-CoV-2. “
Figure 3 legend: The figure shows a representative result from a positive sample amplified using the oligonucleotides described in this study. Two SARS-CoV-2 genomic regions were targeted for validation: the 5'UTR region (228 bp) and the N2 gene region (310 bp). The capillary electrophoresis electropherogram displays two distinct peaks corresponding to the expected fragment sizes, while the 3% agarose gel confirms amplification through the presence of two bands at the expected positions. Lanes that do not show visible bands correspond to negative samples. A 1 kb DNA ladder (Ludwig Biotec) was used as the molecular weight marker.
Figure 4 - the peaks for the same virus should be at the same length in different graphs and directly underneath
Response: The figure shows positive results for some of the viruses included in the diagnostic method presented in this study. In panel 4A, the sample shows a peak corresponding to the expected genomic region of HCoV-OC43, with the primer labeled with the NED fluorophore. In 4B, the detected virus was HPIV-2, labeled with the FAM fluorophore. In 4C, the detected virus was H1N1, also labeled with NED. In 4D, Enterovirus (EV) was detected and labeled with FAM. Finally, in 4E, HMPV-A was identified, labeled with NED. Each electropherogram displays the set of targets labeled with the same fluorophore; however, the sample was subjected to a single-tube reaction containing all oligonucleotides. To present examples of positive samples for different viruses labeled with distinct fluorophores, we omitted from the figure the electropherograms of other targets that did not show additional peaks.
We have also revised and improved the figure legend to enhance clarity and understanding.
Figure 4. Representative electropherograms of positive samples for different respiratory viruses detected by the multiplex MPCR-CE method. (A) HCoV-OC43 (NED-labeled primer); (B) HPIV-2 (FAM); (C) Influenza B virus (H1N1) (NED); (D) Enterovirus (FAM); and (E) HMPV-A (NED). Each panel displays the set of targets labeled with the same fluorophore. All samples underwent a single-tube reaction containing all oligonucleotides.
I would like to express my gratitude to the reviewer for taking the time to analyze our manuscript.
Best regards,
Vivian Alonso Goulart, PhD
Institute of Biotechnology (UFU-Brazil)
e-mail: alonso.goulart@ufu.br
ORCID 0000-0002-2041-0053
Researcher ID C-1121-2017
